# *Stipagrostis pennata* (Trin.) De Winter Artificial Seed Production and Seedlings Multiplication in Temporary Immersion Bioreactors

**DOI:** 10.3390/plants11223122

**Published:** 2022-11-15

**Authors:** Masoumeh Asadi Aghbolaghi, Beata Dedicova, Farzad Sharifzadeh, Mansoor Omidi, Ulrika Egertsdotter

**Affiliations:** 1Department of Agronomy and Plant Breeding, College of Agriculture and Natural Resources, University of Tehran, P.O. Box 4111, Karaj 31587-11167, Iran; 2Department of Forest Genetics and Plant Physiology, Umea Plant Science Center (UPSC), Swedish University of Agricultural Science (SLU), SE-901 83 Umea, Sweden; 3Department of Plant Breeding, Swedish University of Agricultural Sciences (SLU), Alnarp, P.O. Box 190, SE-23422 Lomma, Sweden; 4George W. Woodruff School of Mechanical Engineering, Georgia Institute of Technology, 500 Tenth Street NW, Atlanta, GA 30332, USA

**Keywords:** in vitro cultures, somatic embryogenesis, protein analyses

## Abstract

This study was conducted to develop the protocol for artificial seed production of *Stipagrostis pennata* (Trin.) De Winter via somatic embryo encapsulation as well as test a temporary bioreactor system for germination and seedling growth. Embryogenic calli were encapsulated using sodium alginate and calcium chloride and then sowed in the Murashige and Skoog (MS) germination medium in in vitro cultures. The experiments were conducted as a factorial based on a completely randomized design with three replications. The treatments include three concentrations of sodium alginate (1.5%, 2.5%, and 3.5%), two ion exchange times (20 and 30 min), and two artificial seed germination media (hormone-free MS and MS supplemented with zeatin riboside and L-proline). Germination percentage and number of days needed until the beginning of germination were studied. The highest percentage of artificial seed germination was obtained when 2.5% sodium alginate was used for 30 min (ion exchange time) and when the seeds were placed on the MS germination medium supplemented with zeatin riboside and L-proline. The results of the analysis of variance in the temporary immersion bioreactor system showed that the main effects observed on the seedling growth were associated with different growth hormones in culture media and the number of feeding cycles. Experimental results also indicated that the total protein analyses of zygotic seedlings and seedlings originating from the synthetic seeds showed no statistically significant differences between these samples.

## 1. Introduction

*Stipagrostis pennata* (Trin.) De Winter is one of the valuable fodder grass species growing in desert regions of Iran. It is drought-resistant grass but has a low capacity for seed production, germination, and growth, leaving the species vulnerable to extinction. Through the use of asexual reproduction methods, including somatic embryogenesis and artificial seed technology, *Stipagrostis pennata*’s reproduction can be increased on a large scale to ensure the continuation and further use of the species. *S*. *pennata* is one of the valuable pasture species belonging to the Aristida tribe and the Gramine family. It is also a herbaceous perennial plant with creeping underground stems [1]. This plant is a dominant grass species in the central desert areas of Iran [2]. The genus Stipagrostis Nees in Iran has eight species; three are growing in the desert sand fields (*S. karelini* (Trin. & Rupr) Tzelv; *S. plumosa* (L.) Munro ex T. Anders, and *S. pennata* (Trin.) De Winter) [3].

In addition to Iran, *S. pennata* is also found in the hot deserts of Egypt, Somalia, India, Iraq, Turkmenistan, Afghanistan, Pakistan, Southern Europe, part of Russia, Caucasus, and Western Siberia [4].

The plant shoots and tillers above the soil act as a barrier against the wind flow, and when the wind flow passes through the plants, its dynamic energy is reduced, and hence wind erosion cannot be very effective around grasslands and areas with high plant density (Figure 1A,B).

According to Gong et al. 2009 [5], the adaptability of *S. pennata* species in desert areas indicates that sand-loving plants such as this species have different characteristics in drought conditions, including having narrow leaves, wax coating on leaves, and roots developed. In this species, in addition to the spread and density of the roots, the stems also grow densely on the soil surface, which has an important outcome in reducing wind erosion and carrying sand particles.

Due to its edible and bushy aerial parts, this plant can be introduced as fodder for livestock with low water demand to grow in dry areas. Considering that the natural seeds of the *S. pennata* hardly germinate and grow [6], and it is a species in danger of extinction, propagation techniques through artificial seeds offer a high potential to preserve this species in the natural environment.

In early studies, artificial seeds only referred to encapsulated somatic embryos. Following the use of artificial seeds, shoot tip parts, axillary buds, and stem nodes have also been used as suitable options for the production of somatic embryos and artificial seeds [7,8,9,10,11]. 

An artificial seed is an encapsulated meristem tissue that can convert into a complete plant growing in cultures in vitro or in vivo. Molle [12] successfully used sodium alginate and calcium salt solution to produce artificial carrot seeds, and Salimi [13] induced embryonic callus from *Hyparrhenia hirta* using a combination of 2,4-D and BA and used the resulting embryos in an MS medium [14] containing three percent sodium alginate to produce artificial seeds. After seven weeks of sowing the seeds in the solid medium, half of the seedlings emerged. The highest germination percentage of *Hyparrhenia hirta* artificial seed was when 3% sodium alginate and 75 mM calcium chloride were used for 30 min to encapsulate somatic embryos [15].

Hung and Trueman [16] also investigated different concentrations of sodium alginate and the duration of exposure to calcium chloride in the production of artificial Eucalypt seeds and stated that with 3% sodium alginate and exposure in 100 mM calcium chloride solution for 30 min, the seeds were uniform, round, transparent, and firm. Moradi et al. 2019 [17] observed the effect of alginate matrix composition on the growth rate of artificial seeds obtained by encapsulating somatic embryos in two sunflower hybrids. Plants regenerated from encapsulated somatic embryos were affected by the sodium alginate concentration, the presence or absence of nutrients in the MS culture medium (macro and microelements), sucrose, and plant growth regulators. In this experiment, the presence of nutrients produced stronger seedlings, and the presence of plant growth regulators in the encapsulating matrix caused more branches and leaves as well as a wider root system in the obtained seedlings.

Encapsulated shoot tips obtained from the in vitro propagated sunflower seedlings Moradi et al. 2015 [18] can germinate on an MS hormone-free medium, following the transfer of plants to two commercial substrates containing coco peat and perlite or a mixture of coco peat, perlite, and peat moss. In addition, Tabassum et al. [19] reported 57% germination of artificial cucumber seeds when they were stored at 4 °C for up to 10 weeks.

Unlike in conifer species, where propagation via somatic embryogenesis has been used for a very long time [20] and the advantages and the main obstacles limiting the widespread implementation of bioreactors for the micropropagation of forest trees are currently discussed and reviewed [21], the use of somatic embryogenesis in combination with a temporary immersion system for *S. pennata* practically does not exist. 

That is why the main goal of this study was a protocol developed for the encapsulation of the embryogenic callus, artificial seed production, and germination of *S. pennata* (Trin.) De Winter in combination with the use of a temporary immersion bioreactor system for seedling germination.

## 2. Results and Discussion

There are still unexplored potential methods for the massive propagation of different plant species using non-sexual methods, e.g., artificial seed technologies called synseeds [22] that are derived from totipotent plant cells, tissues, and organs capable of generating complete clonal plants. This technology has been successfully developed and described for many plant species [23].

The present study was conducted to determine the best concentration of encapsulating compounds for artificial seed production and germination for *S. pennata* (Figure 1E–G) as well as to test a temporary immersion bioreactor system for the germination and mass shoot multiplication of this plant species (Figure 1C,D).

### 2.1. Production and Germination of Artificial Seeds

According to the analysis of variance, the percentage of germination and the number of days until germination started were not affected by the triple interaction of the treatments. 

On the other hand, the interaction of the germination medium at the time of ion exchange did not have a significant effect on the percentage of germination, but the main effects of the treatments, the interaction of the germination medium at the concentration of Sodium alginate and the interaction of the concentration of sodium alginate at the time of ion exchange had a significant effect on these growth parameters (Table 1).

**Figure 1 plants-11-03122-f001:**
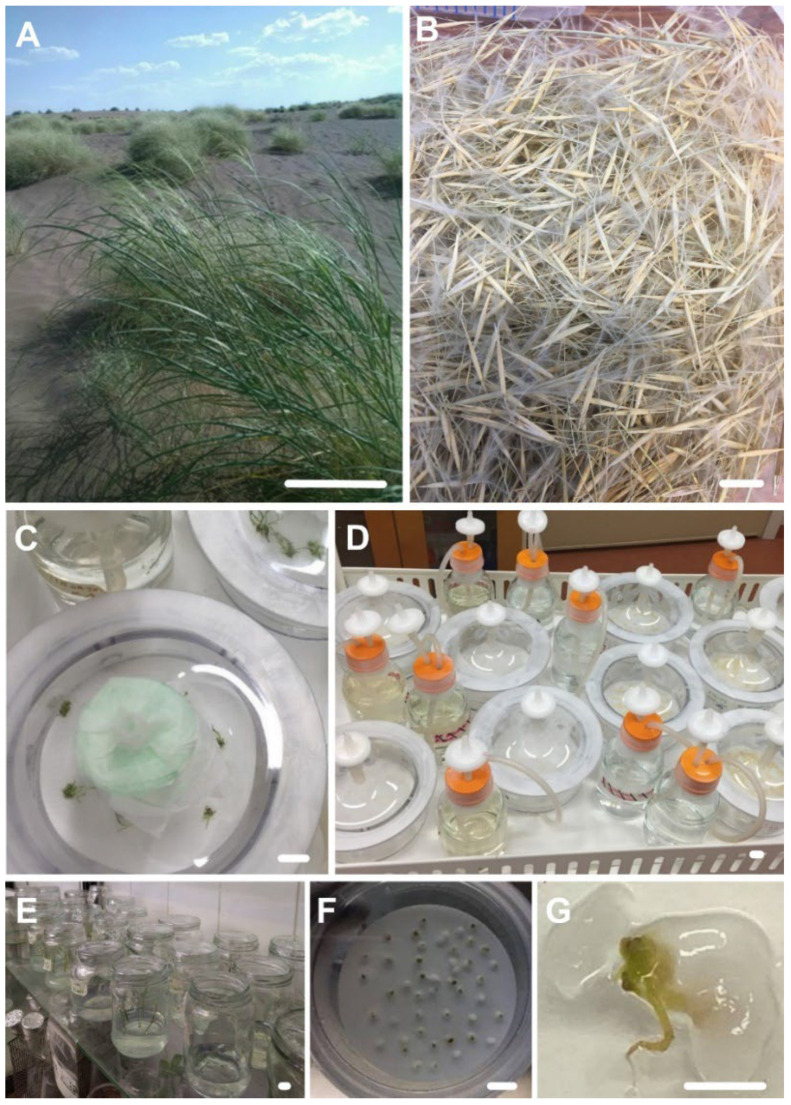
(**A**) *S. pennata* in the desert habitat. (**B**) Seeds of wild growing plants. *(***C**) Temporary immersion bioreactor system with germinated artificial seeds. (**D**) Temporary immersion bioreactor (TIB) system. (**E**) Shoot growth in solid media. (**F**) Artificial seeds of *S. pennata*. (**G**) Germinated artificial seed of *S. pennata*. Scale bar: (**A**) is 10 cm; (**B**–**G**) is 1 cm.

In addition, the main effects of the interaction of the germination medium at the time of ion exchange, the interaction of the germination medium at the concentration of sodium alginate, and the interaction of the concentration of sodium alginate at the time of ion exchange (Table 1) significantly influenced the number of days to germination.

A comparison of the means showed the highest percentage of germination was related to the use of 2.5% sodium alginate in an MS medium containing zeatin riboside and L-proline. In addition, the lowest germination percentage was observed in the MS medium without growth hormone and 3.5% sodium alginate application (Figure 2).

The application of 2.5% sodium alginate and 30 min of ion exchange time increased the germination percentage of artificial seeds, and the highest amount of germination was in the interaction of sodium alginate application at ion exchange time. On the other hand, the use of 3.5% sodium alginate in both ion exchange times caused the lowest germination percentage (Figure 3).

The maximum time until the germination of artificial seeds was obtained in an MS medium without growth hormones with the use of 3.5% sodium alginate (Figure 4), and the use of 2.5% sodium alginate in the medium containing zeatin riboside and L-proline showed the shortest time for germination. The results were statistically similar in the hormone-free medium and the application of 1.5% and 2.5% sodium alginate, and the medium containing zeatin riboside and L-proline and the application of 3.5% sodium alginate were statistically similar.

In the mean comparison of the medium during ion exchange, it was observed that if the medium is free of hormones and the ion exchange time is determined to be 20 min, the maximum time would be required until germination begins (Figure 5). In addition, if the medium contains zeatin riboside and L-proline (no difference in ion exchange time at 20 min and 30 min), the embryo takes the least time to break out of the capsule.

The interaction between sodium alginate concentration and ion exchange time on the number of days until the beginning of germination showed that for the concentrations of 2.5% sodium alginate (with both ion exchange times) and 1.5% sodium alginate (with 30 min of ion exchange), the artificial seed needs the least number of days for germination (Figure 6). On the other hand, the longest time for the embryo to germinate corresponds to the concentration of 3.5% sodium alginate (both times of ion exchange).

The use of sodium alginate is important in the formation of artificial seeds. An increase in sodium alginate concentration causes hardness of the explant protective gel and will hurt the percentage of germination and the time to sprout seedlings from the capsule. This is despite the fact that sodium alginate in very low concentrations does not help to maintain the proper shape and storage of artificial seeds because the formed capsule is very soft and flexible and cannot support the explant well against movement and physical pressure Redenbaugh et al. (1987) [23].

In an experiment by Haque and Ghosh (2016) [24], three concentrations of sodium alginate, 1.5%, 3%, and 4.5%, were used in the preparation of *Ledebouria revoluta* artificial seeds. Even though the highest percentage of germination was obtained in the concentration of 1.5% sodium alginate, it was found that the created capsules were not well polymerized and were not suitable for carrying. A concentration of 3% was considered the optimal condition. On the other hand, the concentration of 4.5% sodium alginate greatly reduced the germination percentage, and the embryo was not able to exit the capsule due to its high hardness. Similar results have been obtained by [25], where sodium alginate’s optimal concentration was 3%.

The research of Inpuay and Te-chato (2012) [26] on oil palm explants showed that the interaction of 2.5% sodium alginate concentration and 15 min of ion exchange time had the best germination percentage of artificial seeds. In addition, the excessive softness of the capsule at a two percent concentration and the high hardness of the gel matrix at a three percent concentration of sodium alginate were stated as reasons for the decrease in germination.

Investigating the interaction of the medium in sodium alginate showed that the highest percentage of germination was in the treatment of a hormone-free medium and a 2.5% concentration of sodium alginate (Figure 2). In addition, the results of the interaction between ion exchange time and sodium alginate concentration showed that the concentration of 2.5% sodium alginate in the ion exchange period of 30 min (Figure 3) considering the treatment combination.

Considering these results and comparing them with the results of other researchers, it seems that the concentration of sodium alginate played a central role in the formation of artificial seeds of *S. pennata*. According to observations, if the appropriate concentration of sodium alginate (both lack and excess concentration) is not present in the gel matrix, with a concentration of 1.5% and the presence of growth hormones, 27% compared to a concentration of 2.5% and the presence of plant growth hormones reduced germination. This amount of reduction was 74% with a concentration of 3.5% and the presence of the plant growth hormone (Figure 2). Similarly, the time of 30 min for ion exchange and the concentration of 2.5% sodium alginate showed a 17% increase in the percentage of germination, which was lower than the statistical level (Figure 3).

According to Ghosh and Sen (1994) [27], the best treatment for artificial seed production was 3.5% sodium alginate and an ion exchange time of 40 min. It seems that the shorter time in the present experiment (30 min) may be due to the synergistic effects of sodium alginate and calcium as well as their concentration. Both sodium alginate concentration and calcium chloride are the key components in the polymerization process, according to Mujib et al. (2014) [28].

Extensive results have been obtained regarding the role of nutrients in the culture environment, including various types of sugars, growth hormones, etc. According to Rizkalla et al. (2012) [29], adding 30 g/L of sucrose to the culture medium for sugar beet explants showed the highest germination percentage, and additionally, the shortest time was needed for starting the germination. Other research also stated that the presence of nutrients in the alginate matrix led to the better regeneration of banana explants (Panis, 1996) [9].

The presence of nutrients in the gel matrix, which are placed around the explant as a nutritional substrate, probably improved the growth and survival of the *S. pennata* explants. We can assume that L-proline is needed to provide the energy for germination. On the other hand, the presence of zeatin riboside stimulated the explant growth and the proper development of mature seedlings.

It was shown by Huda et al. (2007) [30] that sucrose was the most suitable source of carbon supply for eggplant artificial seed production. Similar results were obtained in the production of *Tylophora indica* artificial seeds by Gantait et al. (2017) [31]. The highest percentage of germination and the highest degree of polymerization of the capsule around the explant and, at the same time, the shortest time until the germination occurred when 3% sodium alginate was used.

In general, according to our results, the most suitable conditions for the production of artificial seeds were when 2.5% sodium alginate with a duration of ion exchange of 30 min was used and the artificial seeds were placed on the culture medium supplemented with zeatin riboside and L-proline.

### 2.2. Germination and Seedling Growth on Solid Media and in Temporary Immersion Bioreactors (TIB)

#### 2.2.1. Experiments with Solid Media

All statistical analyses were done according to the PROC GLM method and based on Duncan’s multi-range test. Analysis of variance for micropropagation on a solid culture medium showed that the effects of the plant growth hormones thidiazuron, zeatin riboside, and BA were significantly different only on the growth parameters of the number of stems, biomass, and multiplication rate at the probability level *p* < 0.01. However, there was no significant effect on the length of the longest stem, the percentage of rooted seedlings, and the length of the longest root (Table 2).

According to the mean comparison of the effect of the growth hormone we used in the solid culture medium (Figure 1E), it is clear that thidiazuron produced the highest number of stems, biomass, and multiplication rate. The lowest multiplication rate, biomass, and number of stems on the solid culture medium were related to the medium supplemented with BA (Figure 7).

#### 2.2.2. Experiments in TIB

The results of the analysis of variance in the temporary immersion bioreactor system showed that the main effect of the plant growth hormone and the number of feeding cycles, except for the root-related growth parameters (percentage of rooted samples and the length of the longest root), had a very significant effect on the growth parameters of *S. pennata*. However, the interaction of plant growth hormones in the number of feeding cycles in the TIB system had a significant effect only on the length of the longest stem and did not affect other measured growth characteristics (Table 3).

The medium supplemented with zeatin riboside in combination with a feeding cycle of six times per 24 h showed the highest amount of studied growth parameters of *S. pennata* (Figure 8). By reducing the frequency of the feeding cycle to four per 24 h, the number of stems (Figure 8A), the length of the longest stem (Figure 8B), the multiplication rate (Figure 8C), and the biomass (Figure 8D) showed a significant decrease. The decreasing trend in the mentioned growth parameters continued with the change of the plant growth hormone in the liquid medium of the bioreactor from zeatin riboside to thidiazuron. The greatest decrease was observed in liquid medium supplemented with BA in combination with the feeding cycle four times per 24 h.

In our experiment, the roots of the *S. pennata* seedling showed the smallest effect from the interaction of the number of feeding cycles and the presence of plant growth hormones in the temporary immersion bioreactor system. Therefore, the presence of BA (both feeding cycles) and thidiazuron (four times feeding per 24 h) did not produce any roots, and because of that, the percentage of rooted samples and the length of the longest root was zero. 

However, other growth parameters, including the stem numbers and biomass, as well as the multiplication rate, showed significant changes when two different feeding cycles (4 and 6 times per 24 h) were used in combination with different plant growth hormones in a liquid culture medium (Table 4).

#### 2.2.3. Shoots Micropropagation on Solid Media and in TIB

The final goal of this experiment was to compare the two techniques (solid and liquid culture medium) for the growth and development of shoots. Nevertheless, surprisingly, our results showed a lack of statistical significance of the interaction between plant growth hormones and the number of feeding cycles in the temporary immersion bioreactor (Table 5).

In the continuation of our experiments, a comparison of micropropagation in the solid culture media with micropropagation in the temporary immersion bioreactor system was made in terms of plant growth hormone usage and the two feeding cycles in the bioreactor. This way, according to the obtained results, thidiazuron from the solid culture medium and zeatin riboside with both feeding cycles (4 and 6 h per 24 h) from the temporary immersion bioreactor system were selected and compared. We found that the effects of the treatment on the seedling growth characteristics of the longest stem, biomass, and multiplication rate were significantly different at the *p* < 0.01 level (Table 6). 

According to the comparison of the results of the selected treatments of the solid culture medium and the temporary immersion bioreactor, the change in the type of growth hormone and the type of culture medium did not cause significant changes in the number of stems, the percentage of rooted samples, or the length of the longest root (Table 6). However, the use of a temporary immersion bioreactor system along with the feeding cycle six times per 24 h led to the best results in terms of biomass, multiplication rate, and the length of the longest stem (Figure 9). 

The results indicate that, in general, the use of temporary immersion bioreactors for the micropropagation of *S. pennata* seedlings was more efficient than micropropagation on a solid culture medium.

It is very well known that cytokinins are important plant growth hormones that can stimulate cell division, and their type and concentration in the culture media are effective in increasing or decreasing the rate of plant growth and development Kadota & Niimi (2003) [32].

Ahmadian (2017) [33] reported that thidiazuron can stimulate the lateral meristems in temporary immersion bioreactor for shoot cultures of *Dianthus caryophyllus* (L.), and we also confirmed that the number of produced stems of *S. pennata* was significantly higher in comparison to the other two plant growth hormones we tested. The significant difference in the high multiplication rate of seedlings in the presence of thidiazuron compared to other plant growth hormones (Figure 6) showed the positive effect of this compound in the number of produced stems and, ultimately, the biomass and the multiplication rate as well.

Ghasemiomran et al.’s (2018) [34] research with *Stevia rebaudiana* micropropagation showed that cytokinins used in the culture medium had a significant effect on increasing the number of stems and biomass when 2 mg/L of BA and thidiazuron were used in the culture medium. Unlike BA, which had no effect on root formation in their experiments, we observed no effect on the rooting in our cultures as well.

A comparison of solid and liquid culture media for the micropropagation of potato and *Catharanthus roseus* shoots published by Entisari et al. [35] and Mujib et al. (2014) [28], respectively, showed that potato explants in bioreactors had higher stem numbers, height, and more biomass than in the solid culture medium. The results obtained in the *Catharanthus rozeus* cultures also indicated that the liquid culture medium produced more biomass compared to the solid culture medium, which corresponds with our data as well.

In general, the results of this current study showed that the presence of thidiazuron in combination with a feeding cycle of six times per 24 h in the TIB system is more effective for *S. pennata* seedling growth than growing them on a solid medium.

The advantage of the temporary immersion bioreactors can also be that in this culture system, the seedlings and plants have better contact with the liquid medium, and they were fed through the leaves as well. Nevertheless, the bigger size of the culture vessel allows proper aeration in the space between the seedlings, but on the other hand, the risk of infection and loss of a large number of cultures needs to be taken into consideration when bioreactors are used.

#### 2.2.4. Soluble Protein Analyses

The Bradford method is a fast and fairly accurate method of determining the concentration of an unknown protein in samples. The Bradford assay, a colorimetric protein assay, is based on an absorbance shift of the dye Coomassie brilliant blue G-250 Kruger (2009) [36]. Our analyses of the soluble proteins extracted from the zygotic seedlings (representing the control) and seedlings obtained from the synthetic seeds did not show any statistically significant differences in the volume of proteins in these two different samples, as can be seen in Table 7. Unlike the present study, Ehsanpour and Nejati (2013) [37] and Fathi Rezaei et al. (2020) [38] reported that the amount of protein in the plant tissue from in vitro plants can increase or decrease and likewise can be affected by changes in the composition of the culture media.

## 3. Materials and Methods

### 3.1. Plant Material

Mature and immature caryopses of *Stipagrostis pennata* (Trin.) De Winter were collected from field-grown, self-pollinated plants from two locations in Iran (Khuzestan 31°32′11.5″ N lat 49°03′04.3″ E long); South Khorasan (32°38′16.6″ N lat 59°05′15.7″ E long). The seeds were surface sterilized with 70% ethanol for one min, followed by 2% sodium hypochlorite for 10 min. They were then rinsed with sterile Mili Q water.

For the production of artificial seeds, the embryos induced on an MS culture medium with 3 mg/L 2,4-D were used [39]. Previously, we reported that the best embryogenic callus and shoot induction for *S. pennata* was obtained in MS medium supplemented with 2,4-D, and successful shoot induction was obtained in MS medium supplemented with zeatin riboside [40].

### 3.2. Encapsulation of Embryogenic Callus

In our experiments, we tested three concentrations of sodium alginate, including 1.5%, 2.5%, and 3.5%, two times for ion exchange durations of 20 and 30 min and two different culture mediums (MS medium without hormones and MS medium supplemented with zeatin riboside and L-proline) for the germination of artificial seeds. The experiments were carried out as a factorial experiment based on a completely randomized design with three replications.

For each treatment, 100 mL of autoclaved MS medium, pH of 5.6 to 5.8, was mixed with a filter-sterilized 100 mL of sodium alginate solution of the desired final concentration. Well-developed embryogenic callus cultures were passed through a sterile 500-micrometer sieve and transferred into the prepared endosperm. In this study, MS medium without calcium chloride was considered an endosperm.

The resulting mixture was dripped in 1% calcium chloride solution by a sterile pipette. The calcium chloride solution was already autoclaved and cooled at room temperature. Mixing sodium alginate and calcium chloride causes the exchange of ions, the replacement of sodium ions with calcium ions, and finally, the formation of calcium alginate capsules that surround the embryos.

The treatment time at this stage included 20 min and 30 min of ion exchange time. After these times, the extra ions were washed from the surface of the capsules by rinsing them three times with autoclaved Millipore water. Finally, the excess water was dried using sterile filter paper, and the seeds were placed in a Petri dish for the storage period. The lids of the dishes were closed with parafilm and kept at a temperature of 4 °C.

### 3.3. Effect of Growth Regulators on Artificial Seed Germination and Plant Growth in Solid Cultures

After one month of storage, 25 artificial seeds in each Petri dish were cultured on a solid MS medium supplemented with 2% sucrose and 3 g/L of gelrite (for solidification of the media) in Petri dishes. In this experiment, three different plant growth hormones were tested, 6-benzylaminopurine and thidiazuron in a concentration of 2 mg/L each separately, and zeatin riboside in a concentration of 5 mg/L. The pH of the media was adjusted between 5.6 and 5.8 before autoclaving.

### 3.4. Temporary Immersion Bioreactors (TIB) System

In this study, we also tested the TIB system for the germination of synthetic seeds and shoot growth, where liquid culture media were used in comparison with the solid media in Petri dishes and growth containers. The same system has been used and described for hybrid larch (*Larix* × *eurolepis* Henry) [41].

The temporary immersion bioreactor system was used to investigate the possibility of plant propagation in a liquid environment. In this experiment, we tested MS medium, as in solid cultures (only Gelrite was omitted), in combination with the feeding cycle. The feeding cycle in the TIB was four times or six times per 24 h for 30 s. In each bioreactor, 5 shoot clusters originating from 5 synthetic seeds were placed inside. All components of the solid and liquid media were from Duchefa, Haarlem, The Netherlands.

The culture conditions in the growth chamber were the following: a temperature of 24–25 °C, a light intensity of approximately 50 µmol·m^−2^·s^−1^, and a photoperiod of 16 h of light/8 h of dark (Grow Light Quattro from Venso EcoSolution, Partille, Sweden) for TIB and solid cultures as well.

After two months of cultivation, the average number of stems per synthetic seed, the length of the longest stem, the percentage of rooted seedlings, the length of the longest root, the final biomass, and the multiplication rate (the number of final seedlings divided by the number of initial synthetic seeds) were measured and recorded for three different culture media. The germination of artificial seeds and the number of days until the beginning of germination were determined.

### 3.5. Protein Analyses

Soluble proteins were extracted from in vitro regenerated shoots of *S. pennata* obtained from artificial seeds and seedlings from zygotic seeds, used as a control in this test. For each sample, 30 mg of fresh plant tissue was weighed. Plant tissue was homogenized in 100 µL of extraction buffer (150 mM NaCl, 50 mM Tris-HCl supplemented with 0.2% Triton X-100 pH set an 8.0 followed by the addition of 1 mM protease inhibitor PMS) on ice. The resulting mixture was placed for 30 min at 4 °C, with centrifugation following at 12,000 rpm in a cold centrifuge for 20 min. Supernatants were collected into clean tubes, and soluble proteins were determined by Bradford’s method (1976) [42]. All assays were conducted at 4 °C. Bovine serum albumin (Sigma-Aldrich Co., St. Louis, MO, USA) was used as a standard. The absorbance of the samples was measured at a wavelength of 595 nm by a Multiscan Go spectrophotometer from Thermo Scientific (Waltham, MA, USA).

To prepare Bradford’s reagent according to Kruger (2009) [36], we dissolved 0.1 g of Coomassie Brilliant Blue G250 powder in 50 mL of 96% ethanol, and after stirring for at least one hour gradually, 100 mL of 85% orthophosphoric acid was added, and the final volume was adjusted to 1 L with distilled water. The final solution was filtered and stored in a dark container in the refrigerator.

### 3.6. Statistical Analyses

The experiments were conducted as a factorial based on a completely randomized design with three replications. Statistical analysis was done using SAS 9.2 software (SAS Institute, Inc., Cary, NC, USA). Treatments were grouped using the PROC GLM method based on Duncan’s multi-range test with alpha equal to 0.05%, and graphs were drawn using Excel 2016 software (Microsoft, Redmond, WA, USA). 

## 4. Conclusions

In this research work, we are reporting for the first time a successful encapsulation of *Stipagrostis pennata* embryogenic callus with the following creation of artificial seeds and their germination after a month in dark storage. In general, according to the results obtained in our experiments, the most suitable condition for the production of artificial seeds of the *S. pennata* was the use of 2.5% sodium alginate with the duration of ion exchange of 30 min following the germination of the artificial seeds on the MS culture medium supplemented with zeatin riboside and L-proline in cultures in vitro. Nevertheless, the use of a temporary immersion bioreactor system along with six feeding cycles lasting 30 s during 24 h led to the best results in terms of biomass shoot growth, shoot multiplication rate, and the length of the longest shoot produce compared to the cultures on solid media. The results indicate that, in general, the use of a temporary immersion bioreactor for the shoots micropropagation of *S. pennata* was more efficient than the culture on a solid medium in the traditional containers. However, it still seems necessary to conduct more experiments to especially fine-tune the storage period of synthetic seeds produced (e.g., temperature, humidity, etc.) and the developmental stage of encapsulated somatic embryos in synthetic seeds before germination to increase the germination frequencies with full root development.

## Figures and Tables

**Figure 2 plants-11-03122-f002:**
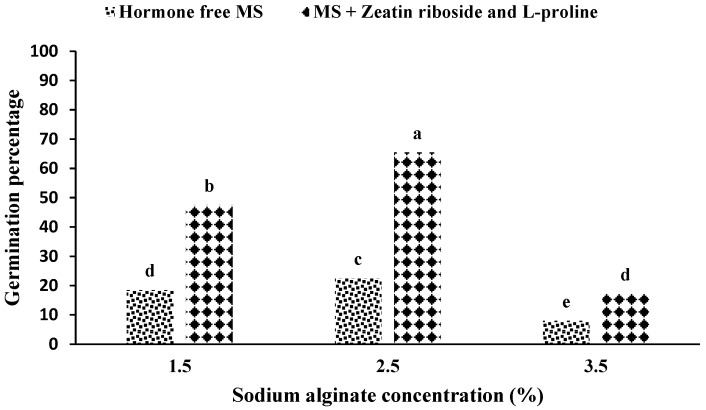
The mean comparison of germination percentage under the influence of germination medium and sodium alginate concentration in artificial seeds of *S. pennata*. Bars with a common letter do not differ significantly from each other. *p* < 0.01 probability level (Duncan’s test).

**Figure 3 plants-11-03122-f003:**
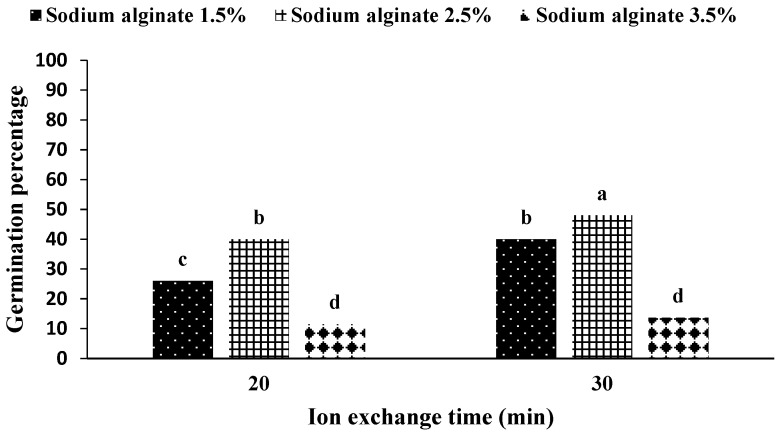
Mean comparison of germination percentages under the influence of ion exchange time and sodium alginate concentration in artificial seeds of *S. pennata*. Bars with common letters do not differ significantly from each other. *p* < 0.01 probability level (Duncan’s test).

**Figure 4 plants-11-03122-f004:**
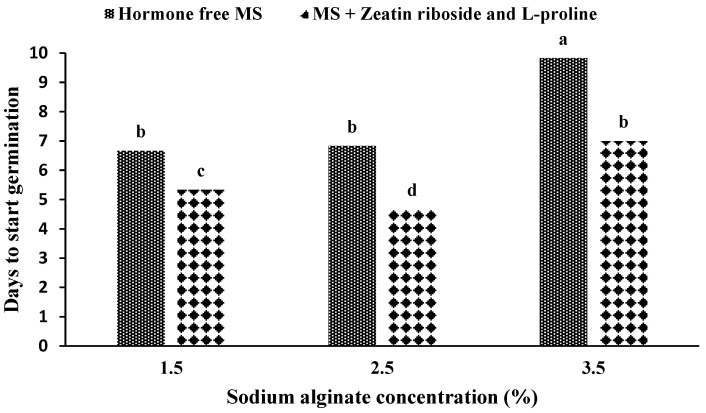
Mean comparison of the number of days until the being of germination under the influence of the germination medium and sodium alginate concentration in artificial seeds of *S. pennata*. Bars with a common letter do not differ significantly from each other. *p* < 0.01 probability level (Duncan’s test). MS—Murashige and Skoog medium.

**Figure 5 plants-11-03122-f005:**
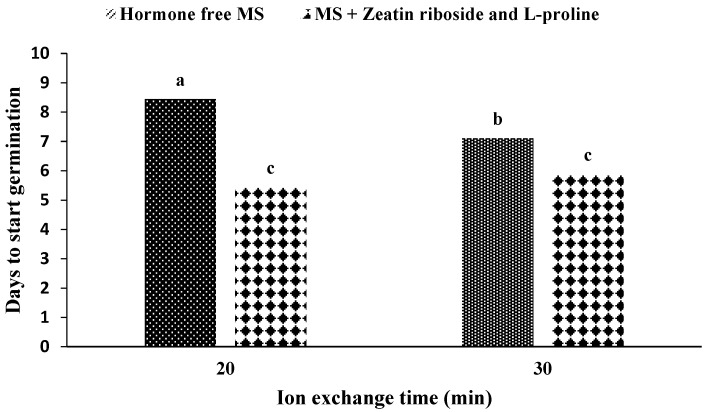
The mean comparison of days until the beginning of germination under the influence of germination medium and ion exchange time in artificial seeds of *S. pennata*. Bars with a common letter do not differ significantly from each other. *p* < 0.01 probability level (Duncan’s test). MS—Murashige and Skoog medium.

**Figure 6 plants-11-03122-f006:**
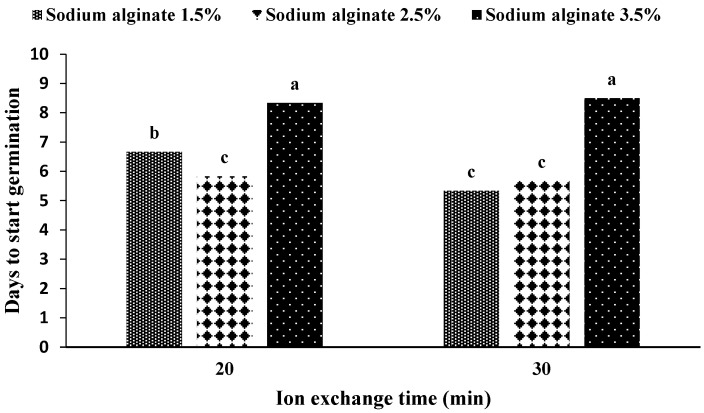
Mean comparison of the number of days until the beginning of germination under the influence of sodium alginate concentration, ion exchange time in artificial seeds of *S. pennata*. Bars with a common letter do not differ significantly from each other. *p* < 0.01 probability level (Duncan’s test).

**Figure 7 plants-11-03122-f007:**
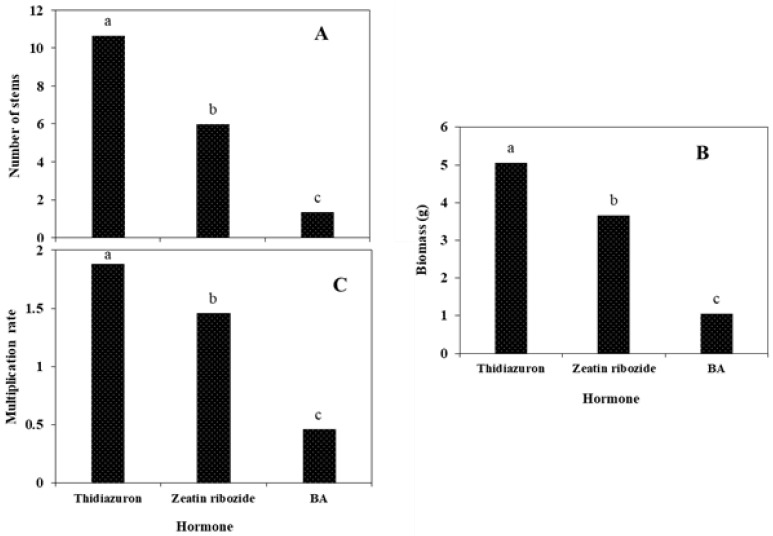
The effect of the plant growth hormones on the number of stems (**A**), biomass (**B**), and multiplication rate (**C**) of *S. pennata* seedlings via micropropagation technique in a solid culture medium. Bars with a common letter are not significantly different from each other. *p* < 0.05 probability level (Duncan’s test).

**Figure 8 plants-11-03122-f008:**
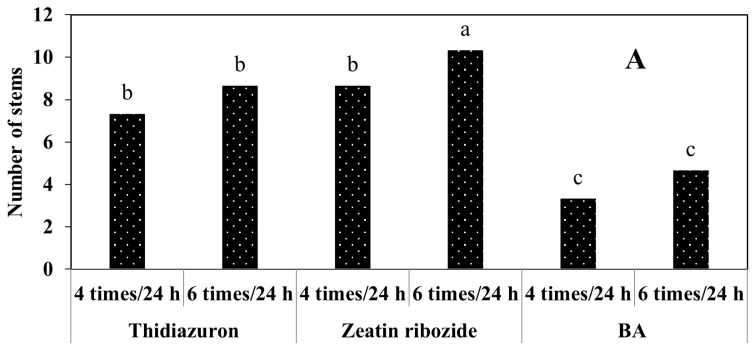
Mean comparison of the number of stems (**A**), the length of the longest stem (**B**), multiplication rate (**C**), and biomass (**D**) of *S. pennata* seedlings under the effect of plant growth hormones and the number of feeding cycles using the temporary immersion bioreactors system. Bars with a common letter do not differ significantly from each other. *p* < 0.01 probability level (Duncan’s test).

**Figure 9 plants-11-03122-f009:**
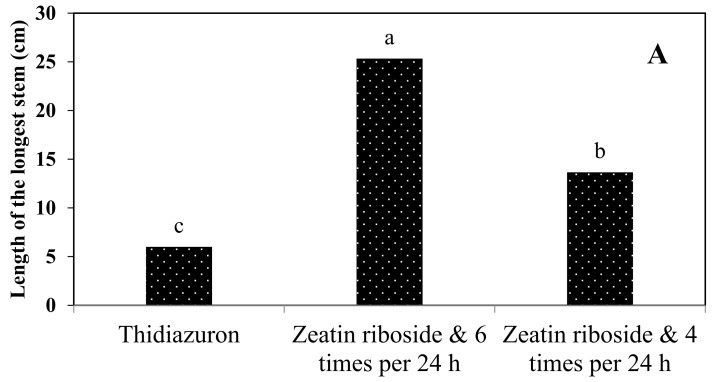
Mean comparison of the effect of thidiazuron (solid culture medium) and zeatin riboside in (TIB) and 4 times per 24 h feeding cycle on the length of the longest stem (**A**), multiplication rate (**B**), and biomass (**C**) of *S. pennata* seedlings. Bars with a common letter do not differ significantly from each other. *p* < 0.01 probability level (Duncan’s test).

**Table 1 plants-11-03122-t001:** Analysis of variance of germination medium effect, sodium alginate concentrations, and ion exchange time on the percentage of germination and the number of days for artificial seed germination in *S. pennata*.

SOV	df	MS	
		Germination (%)	Days to Start Germination
Germination Medium (M)	1	6588.02 **	40.11 **
Sodium alginate (S)	2	3049.69 **	26.02 **
Ion exchange time (T)	1	584.02 **	1.77 *
M × S	2	867.02 **	1.69 **
M × T	1	38.02 ^ns^	7.11 **
S × T	2	105.02 **	1.86 **
M × S × T	2	3.02 ^ns^	0.52 ^ns^
Error	24	9.94	0.25
CV (%)	-	10.5	7.4

^ns^, ** and *, non-significant and significant at *p* < 0.01 probability level and *p* < 0.05 (Duncan’s test). SOV—source of variation; MS—mean square; CV—coefficient of variation; df—degrees of freedom.

**Table 2 plants-11-03122-t002:** Analysis of variance of plant growth hormones affecting *S. pennata* seedling growth characteristics on solid medium.

SOV	df	MS					
		Number of Stems	Length of the Longest Stem (cm)	Rooted Samples (%)	Length of the Longest Root (cm)	Biomass	Multiplication Rate
Hormone	2	3.443 **	0.115 ^ns^	0.012 ^ns^	0.02 ^ns^	1.21 **	0.388 **
Error	6	0.042	0.146	0.025	0.027	0.016	0.003
CV (%)		9.05	14.47	14.73	15.59	7.49	5.41

^ns^ and **, non-significant and significant at the *p* < 0.01 probability level (Duncan’s test). SOV—source of variation; MS—mean square; CV—coefficient of variation; df—degrees of freedom.

**Table 3 plants-11-03122-t003:** Analysis of variance of the plant growth hormone effect and the number of feeding cycles in the TIB system on *S. pennata* seedling growth characteristics.

SOV	df	MS					
		Number of Stems	Length of the Longest Stem (cm)	Rooted Samples (%)	Length of the Longest Root (cm)	Biomass	Multiplication Rate
Hormone (H)	2	1.938 **	4.692 **	0.023 ^ns^	0.155 ^ns^	3.773 **	0.350 **
Feeding cycle (F)	1	0.361 **	2.584 **	0.035 ^ns^	0.133 ^ns^	1.773 **	0.056 **
(H) × (F)	2	0.004 ^ns^	0.385 **	0.012 ^ns^	0.033 ^ns^	0.081 ^ns^	0.011 ^ns^
Error	12	0.023	0.055	0.01	0.052	0.056	0.003
CV (%)		5.81	6.99	9.64	19.91	7.29	4.03

^ns^ and **, non-significant and significant at the *p* < 0.01 probability level (Duncan’s test). SOV—source of variation; MS—mean square; CV—coefficient of variation; df—degrees of freedom.

**Table 4 plants-11-03122-t004:** Effect of plant growth hormone and feeding cycle in temporary immersion bioreactor system on *S. pennata* seedling growth characteristics.

Treatments	Number of Stems	Length of the Longest Stem (cm)	The Number of Rooted Samples (%)	Length of the Longest Root (cm)	Biomass (g)	Multiplication Rate
**Hormone**						
Thidiazuron	8 ^b^	9.5 ^b^	3.33 ^a^	0.31 ^ab^	9.05 ^b^	2.2 ^b^
Zeatin riboside	9.5 ^a^	19.5 ^a^	4 ^a^	0.85 ^a^	17.53 ^a^	3.1 ^a^
Benzylaminopurine	4 ^c^	7.16 ^b^	0 ^a^	0 ^b^	7.05 ^b^	1.63 ^c^
**Feeding Cycle**						
4× in 24 h	6.44 ^b^	9.22 ^b^	0.77 ^a^	0.16 ^a^	9.02 ^b^	2.13 ^b^
6× in 24 h	7.88 ^a^	14.18 ^a^	4.11 ^a^	0.61 ^a^	13.4 ^a^	2.48 ^a^

Mean values with the same letters in each bar and each treatment are not significantly different from each other at the *p* < 0.01 probability level (Duncan’s test).

**Table 5 plants-11-03122-t005:** Analysis of variance of plant growth hormones effect and the number of feeding cycles in the TIB system on the growth characteristics of *S. pennata* seedlings.

SOV	df	MS					
		Number of Stems	Length of the Longest Stem (cm)	Rooted Samples (%)	Length of the Longest Root (cm)	Biomass	Multiplication Rate
Hormone (H)	2	1.938 **	4.692 **	0.023 ^ns^	0.155 ^ns^	3.773 **	0.350 **
Feeding Cycle (F)	1	0.361 **	2.584 **	0.035 ^ns^	0.133 ^ns^	1.773 **	0.056 **
(H) × (F)	2	0.004 ^ns^	0.385 **	0.012 ^ns^	0.033 ^ns^	0.081 ^ns^	0.011 ^ns^
Error	12	0.023	0.055	0.01	0.052	0.056	0.003
CV (%)		5.81	6.99	9.64	19.91	7.29	4.03

^ns^ and **, non-significant and significant at the *p* < 0.01 probability level (Duncan’s test). SOV—source of variation; MS—mean square; CV—coefficient of variation; df—degrees of freedom.

**Table 6 plants-11-03122-t006:** Analysis of variance comparing micropropagation in a solid culture medium with micropropagation in a temporary immersion bioreactor system on the growth characteristics of *S. pennata* seedlings.

SOV	df	MS					
		Number of Stems	Length of the Longest Stem (cm)	Rooted Samples (%)	Length of the Longest Root (cm)	Biomass	Multiplication Rate
Treatment	2	0.091 ^ns^	5.03 **	0.005 ^ns^	0.001 ^ns^	4.23 **	0.186 **
Error	6	0.035	0.117	0.027	0.003	0.066	0.003
CV (%)		6	9.22	14.83	5.29	7.33	3.76

^ns^ and **, non-significant and significant, respectively, at the *p* < 0.01 level (Duncan’s test). SOV—source of variation; MS—mean square; CV—coefficient of variation; df—degree of freedom.

**Table 7 plants-11-03122-t007:** Analysis of variance for soluble proteins obtained from seedlings originating from zygotic and synthetic seeds.

SOV	df	MS
Source of seedling	4	0.057 ^ns^
Error	10	0.026
CV (%)	-	9.2

^ns^, non-significant at *p* < 0.01 probability level (Duncan’s test). SOV—source of variation; MS—mean square; CV—coefficient of variation; df—degrees of freedom.

## Data Availability

Not applicable.

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
