# Peer review of "Stipagrostis pennata (Trin.) De Winter Artificial Seed Production and Seedlings Multiplication in Temporary Immersion Bioreactors"

_plants, 2022, doi:10.3390/plants11223122_

Round 1

Reviewer 1 Report (Previous Reviewer 4)

Dear Authors,

the MS was revised and supplemented with new results that improved the total level of the work. The main existing disadvantage is the lack of data on protein analyses of seedlings, the results of which are mentioned in the conclusion. 

Sincerely,

Reviewer 

Author Response

Reviewer 2 Report (New Reviewer)

Dear Authors,

I started to review the manuscript with interest: "Stipagrostis pennata (Trin.) De Winter Artificial Seed Production and Seedlings Multiplication in Temporary Immersion Bioreactors" by Masoumeh Asadi Aghbolaghi, Beata Dedicova, Farzad Sharifzadeh, Mansoor Omiddi, Ulrika.

The purpose of the research presented in this manuscript was a protocol developed for the encapsulation of embryogenic callus, artificial seed production, and germination of Stipagrostis pennata (Trin.) De Winter.

The Authors have set themselves a noble goal, and the most important thing is that they managed to achieve it, but they did not emphasize this fact in the abstract.

The results of this research undoubtedly deserve publication, but the manuscript still requires a lot of work. Here are some tips to help Authors improve their manuscript:

1) Abstract - the first sentence is suitable for the Introduction section, but redundant here. Please summarize the results obtained in the light of practical application.

2) Keywords - should not repeat words appearing in the title of the manuscript.

3) Introduction - L102-105 - please add a hypothesis.

4) Please arrange the number of references in the text they should start with number 1, 2, 3 and so on ... also order the references accordingly in the References section.

5) Figure 1 - Please explain TIB abbreviation. Please remember that all figures and tables should be understandable without having to look for an explanation of abbreviations in the text.

6) Tables 1, 2, 3, 5 and 6 - please explain abbreviations SOV, MS, CV and df.

7) Figures 2, 4 and 5 - please explain abbreviation MS.

8) Figures 2-8 are very bad quality, not legible, please improve the resolution.

9) Numbering of figures is incorrect, number 6 is on pages 7 and 10.

10) Figure 6 (page 10) - please explain BA abbreviation.

11) Tables 3 and 4 - please explain the TIB abbreviation.

12) Figure 7 - please explain BA and TIB abbreviations.

13) In my opinion, from a practical point of view, the assessment of "germination percentage" is completely useless. Please evaluate the germination capacity in the future, which determines the percentage of normal seedlings that will be allowed to develop further.

14) Conclusion - they should form one compact paragraph. Please work on shortening this section. I believe the results are too detailed there.

15) Author contributions - please prepare in accordance with the template.

16) Overall, the entire manuscript requires a lot of work to conform to the Plants template. Please pay particular attention to the References section.

Overall, I believe the results of the research presented in this manuscript merit publication in the journal Plants.

Round 2

Reviewer 1 Report (Previous Reviewer 4)

Dear Authors,

in my opinion, in its current form, the MS can be published in Plants. 

Sincerely,

Reviewer 

This manuscript is a resubmission of an earlier submission. The following is a list of the peer review reports and author responses from that submission.

Round 1

Reviewer 1 Report

Stipagrostis pennata (Trin.) De Winter is one of the valuable fodder grass species in a desert region, drought resistant but due to the low capacity of seed production, the use of asexual reproduction methods, including somatic embryogenesis and artifical seed technology, can increase its reproduction on a large scale.

Reviewer 2 Report

In this investigation, the development of a protocol for synthetic seed production in Stipagrostis pennata was attempted. Authors evaluated the effect of sodium alginate and ion excange time in synseed formation and two different media to induce seed germination.

The manuscript is poorly written, it includes many sentences which are not precise and would create confusión for international readers: 

Line 37  in the desert areas and the central desert areas of Iran

Line 41 Africa, India, Iraq, Turkmenistan, Afghanistan, Pakistan, Somalia, Arabia, Egypt

Somalia and Egypt are part of Africa

Line 55 different mechanisms in drought conditions, including having narrow leaves, wax coating on leaves, and developed roots

These are not mechanisms but morphological traits

Line 61 very little water requiem?????  What does it mean???

Sentences in lines 44-46   and 56-59 are repetitive

Line 65 Following the use of artificial seeds, shoot tip parts, axillary buds, and stem nodes have also been used as suitable options for the production of somatic embryos and artificial seeds.

This is not correct, shoot tips, axyllary buds and stem nodes have also been used as explants for artificial seed production.

Lines 69-70 Artificial seed is an encapsulated meristem tissue that can transform into the complete plant in vitro or in vivo conditions and maintains this ability. Very poor language

Lines 71-73 are also difficult to understand

Line 74 In general, the goal of artificial seed technology development is to create colonies??????

The rest of the introduction is also poorly structured and written

Results and Discussion

Line 139  On the other hand, the interaction of the germination medium at the time of ion ex change did not have a significant effect on the percentage of germination, but the main effects of the treatments, the interaction of the germination medium at the concentration of sodium alginate and the interaction of the concentration of sodium alginate at the time  of ion exchange had a significant effect on these characteristics (Table 1).

Line 151 In addition, the main effects, are the interaction of the germination medium at the time of ion exchange, the interaction of the germination medium at the concentration of sodium alginate, and the interaction of the concentration of sodium alginate at the time of  ion exchange (Table 1) significantly influence the number of days to germination.  

Both paragraphs are repetitive even though one refers to percentage of germination and the other to the number of days to germination

The rest of this section follows the same trend, repetitive and confusing.

Material And Methods

Line 282 For the production of artificial seeds, the embryos induced on an MS culture medium  with 3 mg/L 2-4-D were used [2].

Authors should briefly describe how the somatic embryos used as explants for seed production had been obtained.

Line 312 The first MS culture medium was hormone-free, following the germination medium supplemented with 5 mg/L of zeatin riboside and 500 mg/L of L-proline added after autoclaving.

Comments indicated above are examples of the way this manuscript has been presented. Based on these comments I recommend the manuscript to be REJECTED.

Reviewer 3 Report

The authors conducted to develop the protocol for artificial seed production of Stipagrostis pennata (Trin.) De Winter via somatic embryo encapsulation. Several comments as following.

1. The introduction should be reorganized. Too many paragraphs in the Introduction. The goals of this research should be added. 

2. It might be clearer that the results and discussions are separately presented. The result should be reorganized, and the title of each result should be added.  

3. The quality of Figures is poor. For example, the bar should be added in Figure 1.

4. The methods should be reorganized.  The title of each experiment should be added.  

Reviewer 4 Report

Dear Authors,

the MS is devoted to an actual problem and provided original data. However, the work was done at a low methodological level. In my opinion, the MS is not suitable for publication in Plants. The main disadvantages of the work: small samples, low quality photos and, most importantly, only visual assessment of germination without molecular genetics (e.g. gene expression analysis) and/or biochemical (e.g. plant hormone analysis) approaches. 

Sincerely